# Machine Learning-Assisted FTIR Analysis of Circulating Extracellular Vesicles for Cancer Liquid Biopsy

**DOI:** 10.3390/jpm12060949

**Published:** 2022-06-10

**Authors:** Riccardo Di Santo, Maria Vaccaro, Sabrina Romanò, Flavio Di Giacinto, Massimiliano Papi, Gian Ludovico Rapaccini, Marco De Spirito, Luca Miele, Umberto Basile, Gabriele Ciasca

**Affiliations:** 1Fondazione Policlinico Universitario “A. Gemelli” IRCCS, 00168 Rome, Italy; maria.vaccaro02@icatt.it (M.V.); flavio.digiacinto@unicatt.it (F.D.G.); massimiliano.papi@unicatt.it (M.P.); gianludovico.rapaccini@unicatt.it (G.L.R.); marco.despirito@unicatt.it (M.D.S.); gabriele.ciasca@unicatt.it (G.C.); 2Dipartimento di Neuroscienze, Sezione di Fisica, Università Cattolica del Sacro Cuore, 00168 Rome, Italy; sabrina.romano@unicatt.it; 3Sezione di Medicina Interna, Dipartimento di Medicina e Chirurgia Traslazionale, Università Cattolica del Sacro Cuore, 00168 Rome, Italy; luca.miele@unicatt.it; 4Dipartimento di Scienze di Laboratorio e Infettivologiche, Fondazione Policlinico Universitario “A. Gemelli” IRCCS, 00168 Rome, Italy; umberto.basile@policlinicogemelli.it

**Keywords:** extracellular vesicles, FTIR, liquid biopsy, cancer, hepatocellular carcinoma

## Abstract

Extracellular vesicles (EVs) are abundantly released into the systemic circulation, where they show remarkable stability and harbor molecular constituents that provide biochemical information about their cells of origin. Due to this characteristic, EVs are attracting increasing attention as a source of circulating biomarkers for cancer liquid biopsy and personalized medicine. Despite this potential, none of the discovered biomarkers has entered the clinical practice so far, and novel approaches for the label-free characterization of EVs are highly demanded. In this regard, Fourier Transform Infrared Spectroscopy (FTIR) has great potential as it provides a quick, reproducible, and informative biomolecular fingerprint of EVs. In this pilot study, we investigated, for the first time in the literature, the capability of FTIR spectroscopy to distinguish between EVs extracted from sera of cancer patients and controls based on their mid-IR spectral response. For this purpose, EV-enriched suspensions were obtained from the serum of patients diagnosed with Hepatocellular Carcinoma (HCC) of nonviral origin and noncancer subjects. Our data point out the presence of statistically significant differences in the integrated intensities of major mid-IR absorption bands, including the carbohydrate and nucleic acids band, the protein amide I and II bands, and the lipid CH stretching band. Additionally, we used Principal Component Analysis combined with Linear Discriminant Analysis (PCA-LDA) for the automated classification of spectral data according to the shape of specific mid-IR spectral signatures. The diagnostic performances of the proposed spectral biomarkers, alone and combined, were evaluated using multivariate logistic regression followed by a Receiving Operator Curve analysis, obtaining large Areas Under the Curve (AUC = 0.91, 95% CI 0.81–1.0). Very interestingly, our analyses suggest that the discussed spectral biomarkers can outperform the classification ability of two widely used circulating HCC markers measured on the same groups of subjects, namely alpha-fetoprotein (AFP), and protein induced by the absence of vitamin K or antagonist-II (PIVKA-II).

## 1. Introduction

In many clinical situations, cancer diagnosis requires single or repeated tissue biopsies of suspected cancerous regions. This procedure is invasive and sometimes associated with pain, discomfort, and risk for the patients. Additionally, the tissue region that needs to be sampled can be hardly or completely inaccessible by interventional procedures, and/or highly heterogeneous, thus leading to ambiguous conclusions. These drawbacks might limit the frequency with which a region can be sampled to check for cancer, thus hindering the possibility to perform an accurate diagnosis, especially in the early stages of the pathology. In this regard, liquid biopsy offers a promising diagnostic alternative, as it relies on the analysis of nonsolid biological tissues, such as blood. As such, it is noninvasive or minimally invasive and can be done more frequently by taking multiple liquid biopsies in a short period, being instrumental for diagnostic purposes and therapy monitoring [1].

Extracellular vesicles (EVs) are lipid-bound vesicles secreted by most cell types into the extracellular space [2,3,4,5,6,7], they abound in bio-fluids and harbor molecular constituents from their parent cells, including proteins, lipids, and genetic material. Due to their easy accessibility and capability of representing their cells of origin, EVs are attracting growing interest as cancer biomarkers in the field of liquid biopsy and personalized medicine. Despite this potential, EVs have still not been widely applied in diagnostics and novel approaches for their characterization are highly demanded.

In this regard, Fourier-transform infrared spectroscopy (FTIR), especially in the Attenuated Total Reflection (ATR) mode, is emerging as a promising tool for label-free molecular profiling of EVs, with several papers published on the subject [4,8,9,10,11,12,13,14,15,16,17,18,19,20]. Most of these works involve in vitro experiments aimed at assessing sample composition and purity [15,16], distinguishing different EV classes [18] or EVs derived from cells under different states/phenotypes/culture conditions [4,8,10,12,21,22,23]. So far, very few FTIR papers have focused on EVs purified from biofluids of subjects enrolled in clinical studies. These papers include Zlotogorski-Hurvitz et al., who studied salivary EVs of patients diagnosed with oral cancer [11], and Yap et al., who characterized EVs purified from the urine of prostate cancer patients [14,24]. To the best of our knowledge, the pioneering paper of Martins et al. on EVs obtained from neurological patients is the only article that uses serum as a biofluid of origin [19], and no studies have been yet published on cancer. Therefore, more data are needed to confirm the capability of FTIR spectroscopy to correctly classify EVs obtained from the serum of patients in different clinical conditions, with emphasis on cancer patients, as well as to test classification algorithms suitable for the high throughput analysis of FTIR spectra of EVs. The latter point is extremely important as EV spectral data can be automatically analyzed with machine learning (ML) methods, giving physicians a direct diagnosis instead of a biophysical parameter that needs further interpretation [8,13,19,20].

This pilot study aims to support the evidence that ATR-FTIR can be used as a fast, reproducible, and effective method for the classification of EV-enriched suspensions purified from sera of cancer patients. For this purpose, samples were extracted from a cohort of 20 patients diagnosed with Hepatocellular Carcinoma (HCC) of nonviral etiology and 19 noncancer subjects. This choice is motivated by the large clinical and research interest in serum biomarkers for HCC screening [25,26,27,28,29,30] and, also, by the growing number of papers reporting potential EV-based HCC biomarkers [31,32]. Here, multivariate statistical analysis and machine learning methods are used for the comparison of mid-IR spectra in the two groups, as schematically represented in Figure 1. Our data highlight the presence of statistically significant differences in the intensity and the shape of several mid-IR spectral bands that could be used as sensitive spectral biomarkers of the pathology. To further stress the relevance of the proposed approach for the development of novel liquid biopsy techniques, we compared the classification performances of our spectral biomarkers with those of serum alpha-fetoprotein (AFP), a glycoprotein produced in early fetal life by the liver and by a variety of tumors, which is currently the most widely used circulating marker in HCC with prognosis and risk of recurrence purposes. Additionally, we measured the serum levels of the protein induced by the absence of vitamin K or antagonist-II (PIVKA-II), a more sensitive circulating marker than AFP for differentiating HCC at all stages in patients with cirrhosis or chronic hepatitis. Notably, a Receiving Operator Curve (ROC) Analysis showed that our spectral biomarkers outperformed both AFP and PIVKA-II in distinguishing the two groups, a result that deserves further studies based on a larger sample size.

## 2. Materials and Methods

### 2.1. Clinical and Laboratory

All the subjects enrolled in this study have given written informed consent. We selected 20 patients with a recent diagnosis of nonviral HCC, according to EASL guidelines [33]. The exclusion criteria were as follows: history of previous treatment (i.e., hepatic resection, liver transplant, anti-angiogenic drugs, and radiofrequency); diagnosis of extra-hepatic neoplasia; HCV or HBV infection; Child-Pugh B or C; obstructive jaundice; and estimated creatinine clearance < 30 mL/min. The inclusion criteria were as follows: Hepatocellular carcinoma at diagnosis; no pharmacological treatment; Child-Pugh A; age > 18. Age- and gender-matched subjects were included in the control group. The exclusion criteria of controls were as follows: evidence of neoplasia; HCV or HBV infection; the presence of monoclonal components; anticoagulant therapy; presence of monoclonal component; and negative inflammatory markers.

Serum samples and clinical data of subjects were processed anonymously. A wide panel of biochemical parameters was measured on the collected blood samples, including Glucose, Cholesterol, Total bilirubin, Albumin, Alanine aminotransferase (ALT), Asand partate aminotransferase (AST), Alkaline phosphatase (ALP), and Gamma-Glutamyl Transferase (GGT). Patients’ sera were collected for quantitative measurement of AFP and PIVKA-II, performed using Lumipulse^®^ G (Fujirebio, Tokyo, Japan), based on Chemiluminescent Enzyme [25,33]. Vascular invasion, portal trunk, and/or main portal branches, were diagnosed by MRI with hepato-specific contrast medium.

### 2.2. Sample Purification

The subjects enrolled in this study underwent peripheral blood sampling and then serum samples were obtained and treated as reported in previous studies [8,13]. EVs were isolated from serum by using the ExoQuick ULTRA precipitation kit (System Biosciences, Palo Alto, CA, USA). The isolation was performed according to the ExoQuick ULTRA manual and within 24 h of the serum sampling from patients. Briefly, blood cell debris was removed from the samples by two sequential centrifugations (3000× *g* for 15 min and 12,000× *g* for 10 min, respectively) and the ExoQuick reagent (67 µL) was added to 250 µL of each clarified biofluid. After 30 min of incubation, final centrifugation was performed and the pellet, which contains precipitated EVs, was resuspended in the ExoQuick buffers and finally purified using the provided columns. The samples were analyzed without any further manipulations. Regarding sample purity, a caveat was necessary. As reported in [34,35], the described purification protocol is likely to introduce contaminants from serum, such as lipoproteins. For this reason, our preparation is more adequately referred to as EV-enriched suspension, rather than a pure EV sample. Due to the presence of such contaminants, we decided not to assign the observed spectral signatures to specific molecular species. Conversely, we preferred to use robust statistical methods to highlight clinically relevant differences between the two groups that can be instrumental to develop novel HCC diagnostic tools. A structural and morphological characterization of the samples extracted with this method is reported in Appendix A.

### 2.3. FTIR Measurements and Data Analysis

The EV-enriched suspensions isolated from sera were analyzed using a Bruker ALPHA II compact FTIR Spectrometer, equipped with an attenuated total reflection (ATR) module (Eco-ATR). According to several papers investigating the spectral characteristics of EVs using ATR-FTIR, sample solution droplets were deposited on the ATR crystal and a thin film was obtained by slow evaporation of the solvent under ambient conditions. Specifically for each measurement, 5 μL of the EV solution was air-dried for at least 10–15 min on a high-throughput ZnSe crystal [4,8,9,10,11,12,13,14,15,16,17,18,19,20]. This step was repeated three times, resulting in a total volume of 15 μL of dried solution. All IR spectra were acquired in the wavelength range included from 4000 to 650 cm^−1^ and, for each spectrum 54 scans, at a resolution of 2 cm^−1^ was averaged. The background was acquired before the measurements and then subtracted from each sample spectra. The spectra were registered and preprocessed using the commercial OPUS 8.5 SP1 software, dedicated to the analysis of IR spectral data. Data were then exported and further analyzed with the programming software R. Spectra were normalized after linear baseline subtraction. Average Spectra were computed together with the corresponding 95% confidence intervals. Data were visualized with the ggplot2 software package. Principal Component Analysis (PCA) and Linear Discriminant Analysis (LDA) of spectra were carried out after subtracting a linear baseline from the analyzed absorption band. This analysis protocol was chosen as PCA alone, as well as PCA-LDA in combination, were successfully exploited in similar papers for discriminating EVs of different origins according to their vibrational fingerprints [8,9,10,11,12,13,14,15,16]. Additionally, PCA-LDA has been widely employed in other FTIR-based classification problems in different fields, including diagnostics and forensic sciences [17,18,19,20,21]. More specifically, PCA is an unsupervised statistical learning technique and was used here to find a low-dimensional representation of spectra that retains as much as possible of the variation in the original dataset [22]. Mathematically, this is achieved by computing the covariance matrix of the whole dataset and finding the matrix eigenvalues with the corresponding eigenvectors. The supervised LDA statistical learning algorithm was then used to maximize the separation between the two groups, a task that is performed by solving an eigenvalue problem similar to PCA but starting from the matrix obtained by dividing the between- and the within-class scatter matrices.

Differences between the two groups were assessed using a Wilcoxon non-parametric test for independent samples. Sensitivity and Specificity values were computed from the confusion matrix and validated using the Leave-One-Out-Cross-Validation (LOOCV) method. In this procedure, the database with the 39 samples is divided into 39 groups. At each algorithm iteration, one subset is selected as the testing set, while the remaining ones are used as the training set. The model is verified by the testing set, thus obtaining a classification rate. The mean of 39 classification rates is calculated and used as a robust statistical estimator [20].

## 3. Results

### 3.1. The Ratio of Different Molecular Classes within EVs Are Different in Cancer Patients and Controls

In this study, we compare the mid-IR spectral response of EV-enriched suspensions obtained from sera of HCC patients and noncancer subjects to search for novel spectral circulating biomarkers. The demographic, clinical, biochemical, and spectral characteristics of the recruited subjects are summarized in Table 1. In Figure 2A,B, two mid-IR absorbance spectra measured on the two groups of subjects are reported. Spectra are computed by averaging the data acquired on EV-enriched suspensions obtained from each of the recruited subjects, separately. For this purpose, absorbance was first normalized after linear baseline subtraction (see Material and Methods), 95% confidence bands (shaded area) are superimposed on the average curves (continuous lines).

At the level of the whole mid-IR range, the average curves look qualitatively resemblant, showing similar spectral signatures. A more in-depth analysis of Figure 2 shows the presence of changes in the spectral shape and in the relative intensity of several absorption bands, namely the carbohydrates and nucleic acids band at 1000–1200 cm^−1^ [16,19], the protein Amide I and II bands [13,36,37], the lipid C-H stretching band at 2800–3000 cm^−1^ [23,38], and the C=O stretching at approximately 1735 cm^−1^, which has been assigned to purine base and ester groups of lipids in EV samples [9,36,37]. These bands are highlighted in green and red color in controls and patients, respectively, and are considered a major source of spectral biomarkers in EV research [4,8,9,10,11,12,13,14,15,16,17,18,19,20,23]. To investigate more in-depth the difference between the two groups, an enlarged detail of the four mentioned IR bands for controls (green) and oncologic patients (red) is reported in Figure 3. Linear baseline subtraction, followed by area normalization, was performed before averaging curves.

A box-plot analysis of the computed areas is reported in Figure 4 for the amide I and II bands at 1470–1700 cm^−1^ (A), the lipid CH stretching band at 2800–3000 cm^−1^ (B), the carbohydrates and nucleic acids band at 1000–1200 cm^−1^ (C), and the C=O stretching peak at 1720–1740 cm^−1^ (D). The result of a Wilcoxon rank–sum test for independent samples was superimposed on each plot. Data are summarized in Table 1. Statistically significant differences in the computed areas were detected for all the studied bands, with emphasis on the CH stretching band that displays a remarkably low p-value. Interestingly, highly significant differences are detected by computing the ratios between the CH-stretching band at 2800–3000 cm^−1^ and the Amide I-II bands at 1470–1700 cm^−1^ (Figure 4E), which is known as the spectral Lipid-to-Protein Ratio (LPR) [8,18], and provide a quantitative index correlated to the relative amount of lipids and proteins in the analyzed samples. Remarkable differences are also observed in Figure 4F that reports the ratio between the CH-stretching band and the absorption peak at 1000–1200 cm^−1^, which can be defined—in analogy to Figure 4E—as the spectral Lipid-to-Nucleic acids Ratio (LNR). Taken together, Figure 4E,F hint at the occurrence of an alteration in the ratio between different molecular classes in EVs obtained from cancer patients compared to controls.

### 3.2. Machine Learning-Assisted Classification of HCC Patients and Controls

Aside from intensity values, differences in the average line shape between the two groups can be observed for all the IR bands reported in Figure 3, except for the carbohydrates and nucleic acid band. In Figure 5A–I, we investigate the possibility to use multivariate statistical and Machine Learning methods to automatically classify patients according to their group membership based on the spectral shape of the analyzed bands. For this purpose, we show the projections of the spectral data in the plane of the two principal components (PC1 and PC2) for the amide I–II region (A), the CH-stretching band of lipids (D), and the CO stretching peak at 1720–1760 cm^−1^ (G). We then used Linear Discriminant Analysis (LDA) to point out the direction (LD1) that maximizes the separation between the two groups, which is represented as a continuous black line together with the corresponding datapoints projection. The LD1 discriminant scores for the three analyzed bands are reported in Figure 5B,E,H (lower panels), respectively. A sufficient degree of clustering among data points belonging to the two groups can be observed, as confirmed by the corresponding box-plot analysis (upper panels), which points out the presence of statistically significant differences. In this context, remarkably low p-values are measured for the IR band analyzed in Figure 5H (*p* = 4.9 × 10^−5^). The PCA-LDA classification performances were tested with the leave-one-out-cross validation method (LOOCV) and the results are visualized in Figure 5C,F,I, respectively. These results are extremely interesting if compared with those obtained on conventional HCC circulating markers, such as AFP, which shows a sensitivity of 0.61 (95% CI 0.60–0.62) and a specificity of 0.86 (95% CI 0.85–0.87) in the 20–100 ng/mL concentration range, as demonstrated by a recent metanalysis [28].

### 3.3. A Combined Spectral Biomarker for HCC Diagnosis

In Figure 6, we evaluate the classification performance of each of the spectral biomarkers highlighted in Figure 4 and Figure 5 using receiving operator characteristics (ROC) curves and computing the corresponding Areas Under the Curve (AUC). The same analysis is carried out for two widely studied HCC circulating markers, namely AFP and PIVKA-II. The corresponding AUC values are summarized in Table 2, together with the corresponding confidence intervals. Interestingly enough, all the measured AUCs are statistically different from the random classifier (AUC = 0.5), which is represented by the continuous black line in Figure 6A. In Figure 6B, we investigate the possibility to use biomarkers in combination to improve the accuracy of the classification. For this purpose, we performed a multivariate logistic regression including all the markers in Table 2, followed by a stepwise logistic regression aimed at selecting the most informative set of parameters. This procedure selects two spectral biomarkers that can be used in combination, namely the integrated area of the CH stretching band at 2800–3000 cm^−1^ and the LD1 coefficients computed for the spectral signature at 1720–1760 cm^−1^. The result of the logistic regression performed on the stepwise model shows that the two selected markers possess statistically significant regression coefficients. The ROC curve of the stepwise model is reported in red in Figure 6B, showing a very large area under the curve (0.91, 95% CI: 0.8–1), which confirms the diagnostic potential of the combined spectral biomarker. Youden’s method was used to compute the sensitivity and specificity starting from the combined ROC curve and obtaining 1.00 and 0.76, respectively. ROC curves obtained on the same groups of subjects for AFP (gold curve) and PIVKA (violet curve) are reported for comparison in Figure 6B, obtaining lower—albeit comparable—AUC values, namely 0.81 and 0.86, respectively discussed.

## 4. Discussion

In this paper, we discuss the proof of concept of a novel liquid biopsy approach for cancer diagnosis. Our method is based on the molecular profiling of EVs with FTIR spectroscopy. For this purpose, EV-enriched suspensions were obtained from the serum of patients diagnosed with HCC of nonviral origin and noncancer subjects. We have chosen to investigate EVs because of their crucial role in the emerging liquid biopsies for cancer diagnosis [39,40,41,42,43]. Such a pivotal role is due to their large availability in biofluids and specific molecular content, which reflects the state of the cells of origin. Additionally, EV-specific molecular cargoes change during cancer evolution, thus being also instrumental for disease staging [44]. As far as HCC is concerned, EVs play a particularly relevant role in the disease progression, as cancer cells are capable of influencing many biological pathways through the release of extracellular vesicles in the tumor microenvironment [32]. These effects include a local regulation of the Epithelial to Mesenchymal Transition (EMT) [45], and the activation of the inflammatory microenvironment to increase cancer cell invasiveness, converting fibroblasts and macrophages to CAFs and TAMs [32,46,47]. Moreover, HCC-related EVs are believed to regulate the functions of immune and endothelial cells, inducing immune escape and angiogenesis [48,49].

HCC diagnosis is currently carried out by combining different types of information, which include the levels of selected circulating markers, medical imaging (US/MRI/CT), and histopathological biopsies [25,33,50]. Unfortunately, all these methods present some limitations, which negatively affect patient prognosis. Circulating markers, AFP included, have usually low sensitivity and specificity, although improved performance can be obtained through the combination of multiple markers [25,26,27,29,33]. Despite a remarkable specificity, medical imaging has a poor capability of detecting small tumors, being plagued by a large number of false negatives (FNs) and, consequently, a poor sensitivity. Histopathological biopsies also present several problems such as their invasiveness and a large number of FNs, which are partly due to the intrinsic variability of the bioptic tissue that complicates the sampling of the suspected region [25,32,33,51,52]. EV-based liquid biopsy can contribute to overcoming some of these shortcomings, as it is non- or minimally invasive and can provide a homogeneous sampling. Here, we used multivariate statistical analyses and Machine Learning (ML) methods to search for IR spectral biomarkers of the disease. We highlighted relevant differences in, both, the integrated intensity (Figure 4) and the line shape (Figure 5) of major mid-IR absorption bands of biomolecules within EVs. Specifically, HCC patients presented with a decreased intensity of the amide I and II bands (1470–1700 cm^−1^) and the nucleic acid and carbohydrate band (1000–1200 cm^−1^) compared to the controls, together with an increased intensity of the lipid CH stretching band (2800–3000 cm^−1^). These findings hint at a difference in the ratio of molecular classes within EVs in the two groups, as confirmed by the measured spectral Lipid to Protein ratio (LPR, Figure 4E) and the Lipid to carbohydrates and Nucleic acid Ratio (Figure 4F). This point deserves a more in-depth discussion as it helps point out the complementarity of FTIR with respect to conventional omics techniques for the biochemical characterization of EVs. Genomics, proteomics, and lipidomics, indeed, provide detailed quantitative information on the EV molecular cargos but involve complex sampling procedures that can alter the ratio among different molecular classes. On the contrary, FTIR spectroscopy appears to be perfectly suited to provide semiquantitative information on the relative amount of lipids, proteins, DNA, and RNA in EVs, also highlighting possible biochemical changes that depend on the clinical conditions of patients [8,9]. Investigating quantitative parameters, such as the shape or intensity of spectra, is a reasonable approach, as long as each sample has been uniformly processed. With this idea in mind, several works applied ML algorithms for the analysis of several circulating biomarkers and the development of novel cancer detection methods [11,53,54,55,56,57,58]

In this context, the same LPR here exploited has been also used in the pioneering paper of Mihàly et al. for discriminating among different classes of EVs [18] and in Romanò et al. to classify EVs obtained from cancer cells in different stages of the epithelial to mesenchymal transition [8].

Aside from the relative bands’ intensity, we found statistically significant differences in the shape of three specific mid-IR bands encompassing the spectral ranges 1470–1700 cm^−1^ (Figure 5A), 1720–1760 cm^−1^ (Figure 5B), and 2800–3000 cm^−1^ (Figure 5C). These differences were pointed out using ML, thus allowing us to automatically classify patients according to their clinical conditions. We believe that this approach is highly interesting in a clinical setting as it is designed to provide physicians with a direct diagnostic response instead of a set of spectral parameters that need to be interpreted by specialized personnel. Additionally, compared to IR peak assignment, the use of ML for spectral classification has the potential to reduce the detrimental effects due to the presence of non-EV contaminants in purified samples, a well-known problem for vibrational spectroscopy of EV samples and associated with the lack of a reproducible gold standard purification method to isolate EVs from serum with high yield and purity [4,59,60,61]. For instance, Lee et al. investigated SERS spectra from EVs purified with several isolation techniques and concluded that commercial kit reagents had a strong affinity to EVs leading to relevant coprecipitation [62]. Accordingly, Pereira et al. showed that a significant portion of the IR spectral signals of EVs isolated from cell culture media is ascribed to kit reagents [10]. Therefore, coprecipitants from the isolation kits along with serum proteins, such as albumin, lipoproteins, or immunoglobulins, are potential contaminants that may affect the downstream analyses of EVs. In such a complex condition, the IR peak assignment is extremely challenging and potentially misleading, while the results of an ML-based classification are likely to be more robust, as the interaction between EV and contaminants, which may lead to coprecipitation, may be different in different types of samples. This is especially true if one considers recent findings demonstrating the formation of a protein corona around extracellular vesicles dissolved in complex media [63].

EV-based HCC biomarkers were reviewed in the excellent papers of Wang and colleagues [32] and Szabo & Momen-Heravi [64], which showed that microRNAs, long noncoding RNAs (lncRNAs), mRNAs, circular RNAs, and proteins within EVs are promising sources of HCC biomarkers for diagnosis and staging. MicroRNAs are highly abundant in EVs and difficult to be degraded, thus being an ideal biomarker candidate. Altered microRNA levels in HCC patients have been measured for EV miR-9-3p[65], miR-21[66], miR-93[67], miR-92b [68], miR-718 [69,70], miR-122 [71,72], miR-638 [73,74,75], and miR-125b [76], to mention a few. Altered EV lncRNA (LINC00161[77], lncRNA-HEIH [78,79], ENSG00000258332.1, and LINC000635[80], lncRNA-ATB[81]), mRNA (mRNA-hnRNPH1), circular RNA (circPTGR1[82], and circDB [83]) levels were also detected in HCC patients compared to controls. EV proteins are also important markers in the diagnosis of HCC and alterations in the levels of several proteins have been detected including, G3BP, PIGR [84], hepcidin [85], and SMAD3 protein[86]. Despite the promising diagnostic role of the mentioned RNA and protein structures, clinical trials have pointed out that a single biomarker at a time display low sensitivity and specificity. Comprehensive detection and evaluation using multiple molecules in combination may be more effective in diagnostics [25,32,33]. In this context, the spectral differences highlighted in the intensity of the nucleic acid and carbohydrates band and the shape and intensity of the amide I and II bands are highly interesting, as the former is associated with nucleic acids [19,87], while the latter with the vibrational modes of proteins, but also with the in-plane vibrations of nitrogenous bases of RNA [9,18,88,89]. Therefore, an alteration in the levels of the mentioned protein and RNA markers is likely to affect the shape and the intensities of the analyzed mid-IR bands, producing detectable spectral changes (Figure 5). Additionally, we found highly significant differences in the shape of the lipid-ester peak at about 1730 cm^−1^ and in the lipid stretching band at 2800–3000 cm^−1^. This is an interesting finding and confirms that—aside from RNA and protein markers—the EV lipid content is an effective source of HCC markers, as supported by the fact that HCC development and progression have been linked to fatty acid metabolism dysregulation, in which aberrant activation of oncogenic signaling pathways alters the expression and activity of lipid-metabolizing enzymes [90]. 

Finally, in Figure 5 and Figure 6, we evaluated the classification performances of our spectral biomarkers alone and in combination using the LOOCV method and a ROC curve analysis followed by AUC calculation. The obtained results are interesting if compared with those obtained on conventional HCC circulating markers, i.e., AFP and PIVKA-II, in the same groups of subjects (Figure 6B,C). Despite that AFP is currently the most widely used serum HCC biomarker, its sensitivity at low cut-off values (less than 20 ng/mL) is approximately 60%, and the specificity is still inadequate [91,92]. Furthermore, serum AFP levels remain normal in 15–30% of advanced HCC [93], as, in our case, stressing the importance of finding other circulating markers capable of detecting AFP-negative cancer patients. In this context, PIVKA-II, Protein Induced by Vitamin K deficiency or antagonist-II (PIVKA-II), is believed to be a favorable biomarker to detect AFP-negative HCC [27,29,94,95]. Figure 6B,C show that our combined spectral biomarker possesses a larger, albeit comparable, AUC value than AFP and PIVKA-II measured on the same subjects. Very interestingly, if the sensitivity is computed from the ROC curves using the Youden’s method, we obtained a 70% sensitivity for AFP, 85% for PIVKA-II and 100% for our combined spectral biomarker. A similar outcome was obtained with the “closest top-left” method for computing sensitivity. Additionally, since no significant correlation was found between spectral markers and the levels of PIVKA-II and AFP (data not shown), it would be interesting to investigate whether the formers can contribute to the detection of HCC-negative cancer patients.

A limitation of this study is the reduced sample size, which is a common bottleneck affecting the vast majority of the published FTIR study on this subject. Therefore, the results here presented need to be validated on a larger sample size, which could serve also as a platform to test other machine learning algorithm than PCA-LDA, such as PLS-DA, which have been shown to outperform PCA-LDA on large datasets [96,97].

## 5. Conclusions

In this work, we present, for the first time in the literature, the proof-of-concept of a liquid biopsy approach for cancer diagnosis based on the molecular profiling of serum EVs with FTIR spectroscopy. A cohort of HCC patients of nonviral etiology was selected as a model system and compared to a cohort of noncancer subjects. As deeper argued in the discussion section, the alteration of the molecular content in EVs isolated from HCC patients has been reported by several excellent works. While the study of these biochemical changes at the single-molecule level is needed to have a better understanding of tumor biology, the combination of multiple circulating markers is highly demanded to improve diagnostic performances in a clinical setting. In this context, FTIR spectroscopy appears to be extremely promising as it can provide a label-free comprehensive molecular fingerprint of EVs through the analysis of specific mid-IR absorption bands. Here, we analyze the FTIR spectra of serum EVs by applying multivariate statistics and Machine Learning to highlight clinically relevant variations between the two groups. Our data point out the presence of statistically significant differences in the integrated intensities of major mid-IR absorption bands, including the carbohydrate and nucleic acids band, the protein amide I and II bands, and the lipid CH stretching band. These findings are efficiently captured by the lipid-to-protein spectral ratio (LPR, Figure 4E) and the Lipid-to-Nucleic Acids spectral ratio (LNR, Figure 4F), which show that HCC-derived EVs have an altered balance among the levels of different molecular classes compared to the controls. Additionally, we used PCA-LDA for the automated classification of spectral data according to the line shape of the mentioned mid-IR spectral bands. This analysis identified three potential spectral biomarkers for HCC diagnosis, namely the shape of the Amide I and II bands, the lipid CH stretching band, and the lipid-ester peak at about 1740 cm^−1^. The diagnostic performances of all the proposed spectral biomarkers, alone and in combination, were evaluated by ROC curve analysis. A stepwise multivariate logistic regression allowed us to select the most informative subset of spectral biomarkers suitable for patient classification. The combined biomarker displayed good classification potential (Figure 6), as confirmed by a large AUC value (0.91) and a sensitivity and specificity of 1 and 0.76, respectively. Very interestingly, our combined spectral biomarker appears to outperform the classification performance of two widely used circulating HCC markers—namely, AFP and PIVKA-II—measured on the same group of subjects, a result that deserves a dedicated study based on a larger sample size.

## Figures and Tables

**Figure 1 jpm-12-00949-f001:**
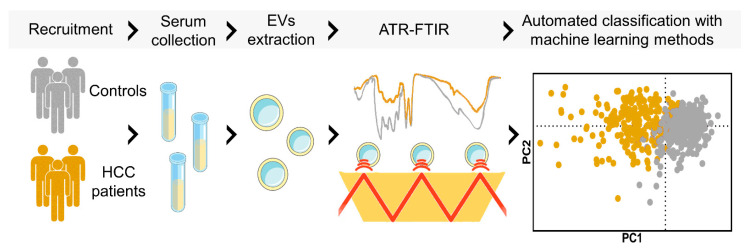
Schematic workflow of the presented EV-based liquid biopsy. Briefly, serum samples from HCC patients and controls were collected. EVs were isolated from serum samples and ATR-FTIR measurements were performed. Finally, automated classification of the two groups was achieved by machine learning analysis of the acquired spectra.

**Figure 2 jpm-12-00949-f002:**
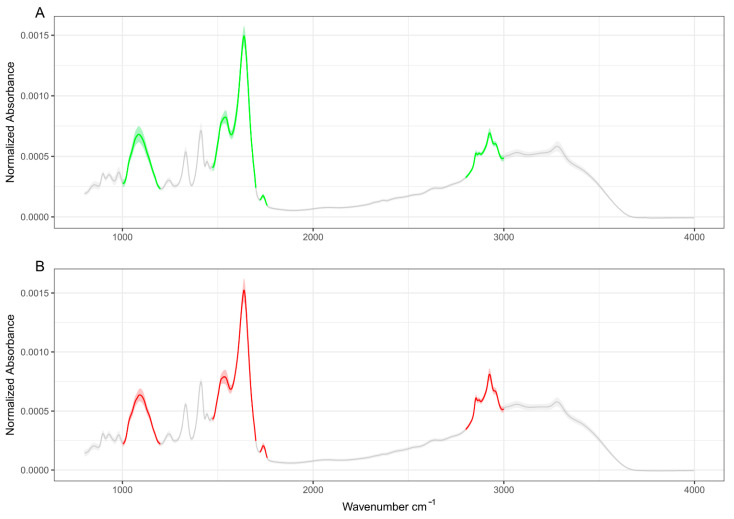
Average absorbance spectra of control subjects (**A**) and HCC patients (**B**). Data are reported as mean ± 95% confidence bands. The analysis performed in this study is principally focused on the highlighted green (controls) and red (patients) bands.

**Figure 3 jpm-12-00949-f003:**
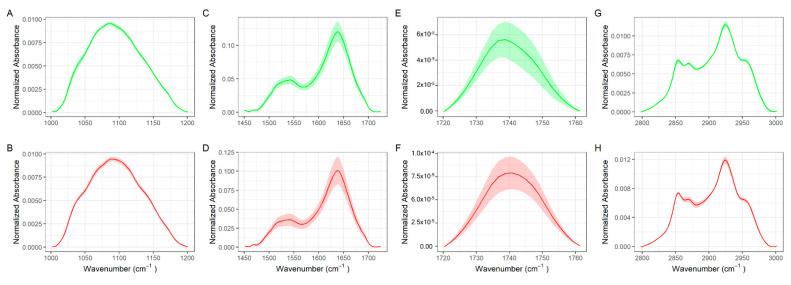
Enlarged details of characteristics absorption bands of biomolecules in EVs in control (green) and HCC patients (red), namely the carbohydrates and nucleic acids band at 1000–1200 cm^−1^ (**A**,**B**), the amide I and II bands at 1470–1700 cm^−1^ (**C**,**D**), the C=O stretching band at 1720–1740 cm^−1^ (**E**,**F**), and the lipid CH stretching band at 2800–3000 cm^−1^ (**G,H**).

**Figure 4 jpm-12-00949-f004:**
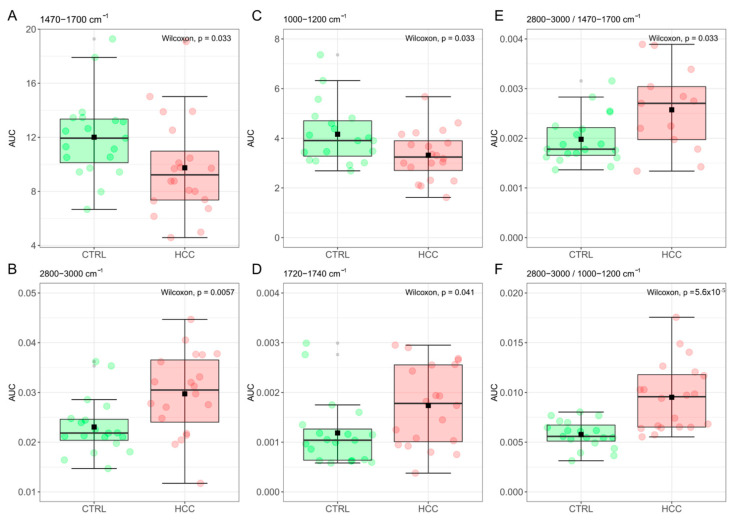
Box-plot analysis of areas computed for the amide I and II bands at 1470–1700 cm^−1^ (**A**), the lipid CH stretching band at 2800–3000 cm^−1^ (**B**), the carbohydrates and nucleic acids band at 1000–1200 cm^−1^ (**C**), and the C=O stretching band at 1720–1740 cm^−1^ (**D**). The ratio between the integrated area computed in panels B and A (**E**) and B and C (**F**).

**Figure 5 jpm-12-00949-f005:**
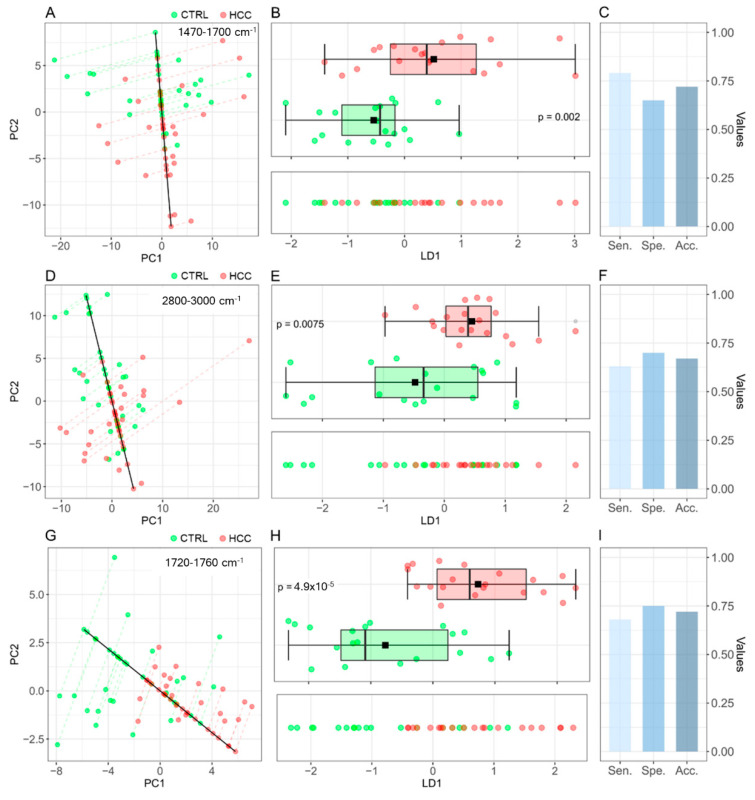
(**A**) Projections of the spectral data in the PC1–PC2 plane for the region enclosed between 1470–1700 cm^−1^, together with the LD1 line (black continuous line), which maximizes the separation between controls (green points) and HCC patients (red points). (**B**) LD1 discriminant scores from the region enclosed between 1470 and 1700 cm^−1^ are reported in the lower panel. The presence of statistically significant difference between the LD1 scores of the two groups is supported by a box-plot analysis (upper panel). (**C**) The PCA-LDA (1470–1700 cm^−1^) classification performances are reported in terms of sensitivity, sensibility, and accuracy. (**D**–**F**) The same analyses were performed for the region enclosed between 2800–3000 cm^−1^ and (**G**–**I**) the region enclosed between 1720–1760 cm^−1^.

**Figure 6 jpm-12-00949-f006:**
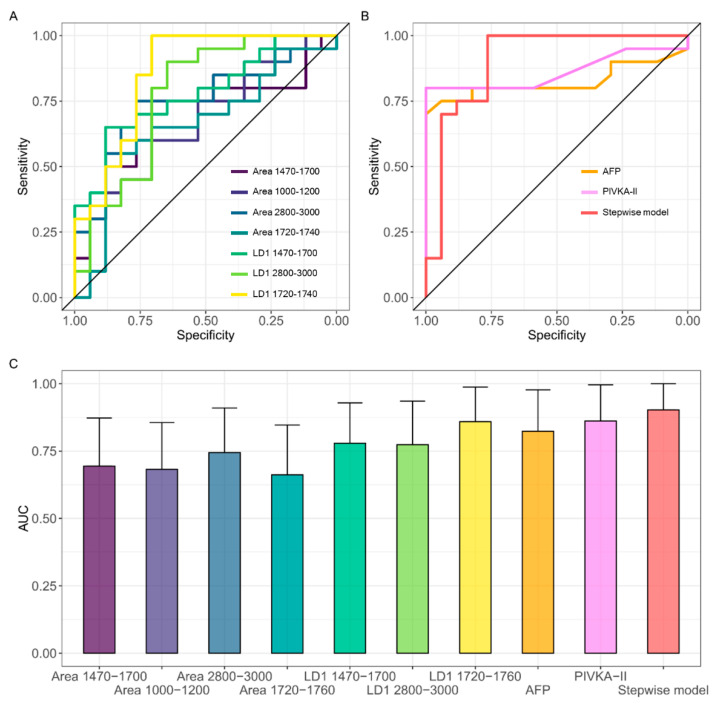
ROC analysis of the all the analyzed spectral biomarkers (**A**) and of AFP (gold), PIVKA-II (violet), and of the combined spectral biomarkers obtained with stepwise logistic regression (red) (**B**); AUC values for each of the investigated markers, alone and in combination, together with the corresponding 95% confidence interval (**C**). The same color scale is used in the three panels.

**Table 1 jpm-12-00949-t001:** Demographical, clinical, biochemical, and spectral characteristics of the recruited subjects.

Characteristic	CTRL, N = 19 ^1^	HCC, N = 20 ^1^	*p*-Value *^2^*
Age (years)	65.8 (4.8)	68.7 (5.9)	0.14
Gender			0.7
F	26%	20%	
M	74%	80%	
Cholesterol (mg/dL)	180 (19)	170 (46)	0.5
Triglycerides (mg/dL)	122 (38)	101 (30)	0.023
GPT-ALT (UI/L)	15 (3)	45 (23)	<0.001
GOT-AST (UI/L)	20 (5)	47 (21)	<0.001
AST/ALT	1.43 (0.53)	3.86 (11.37)	0.3
GGT (UI/L)	31 (5)	179 (147)	<0.001
ALP (UI/L)	171 (17)	272 (137)	0.049
Hb (mmol/L)	14.32 (0.96)	12.78 (2.78)	0.15
Creatinine (mg/dL)	0.79 (0.18)	1.42 (1.29)	0.071
Azotemia (mg/dL)	15 (3)	21 (10)	0.067
Bilirubin (mg/dL)	0.69 (0.23)	1.63 (1.04)	<0.001
Area (1470–1700 cm^−1^)	12.0 (3.1)	9.7 (3.6)	0.033
Area (1000–1200 cm^−1^)	4.16 (1.22)	3.32 (0.99)	0.033
Area (2800–3000 cm^−1^)	0.023 (0.006)	0.030 (0.008)	0.006
Area (1720–1760 cm^−1^)	0.0012 (0.0007)	0.0017 (0.0008)	0.041
LD1 (1470–1700 cm^−1^)	−0.54 (0.77)	0.52 (1.18)	0.002
LD1 (2800–3000 cm^−1^)	−0.48 (1.24)	0.46 (0.70)	0.0075
LD1 (1720–1760 cm^−1^)	−0.77 (1.10)	0.73 (0.90)	<0.001
PIVKA	14 (4)	3205 (9987)	<0.001
AFP	2 (1)	193 (417)	0.001

^1^ Mean (SD); %. ^2^ Wilcoxon rank–sum test; Fisher’s exact test.

**Table 2 jpm-12-00949-t002:** ROC-AUC Table analysis of the significant spectral biomarkers highlighted in Figure 4 and Figure 5. Conventional HCC circulating biomarkers—namely AFP and PIVKA-II—are also included in the analysis together with the results of a stepwise combined model selecting the most informative biomarkers for the classification of the two groups.

Variable	AUC	95% CI
Area (1470–1700 cm^−1^)	0.700	0.524–0.876
Area (1000–1200 cm^−1^)	0.700	0.534–0.866
Area (2800–3000 cm^−1^)	0.755	0.596–0.915
Area (1720–1760 cm^−1^)	0.692	0.518–0.866
LD1 (1470–1700 cm^−1^)	0.776	0.627–0.926
LD1 (2800–3000 cm^−1^)	0.708	0.534–0.881
LD1 (1720–1760 cm^−1^)	0.826	0.694–0.959
PIVKA–II	0.805	0.649–0.961
AFP	0.862	0.728–0.996
Stepwise model *	0.910	0.803–1

* LD1 (1720–1760 cm^−1^) + Area (2800–3000 cm^−1^).

## Data Availability

The data presented in this study are available upon reasonable request to the corresponding author.

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
