# Peer review of "Machine Learning-Assisted FTIR Analysis of Circulating Extracellular Vesicles for Cancer Liquid Biopsy"

_jpm, 2022, doi:10.3390/jpm12060949_

Round 1
Reviewer 1 Report
The paper entitled “Machine learning-assisted FTIR analysis of circulating extracellular vesicles for cancer liquid biopsy” from R. Di Santo et al. describes the implementation of FTIR, coupled with machine learning-driven interpretation of experimental data to the detection of HCC from human sera.
The approach described by the authors is fascinating and well described, and I am particularly impressed by the implications in the diagnostic field. The use of FTIR surely represents an easy-to-implement approach with exceptional advantages such as low sample volume and short analytical time.
Nevertheless, I believe that a major revision is needed in order to publish this work.
In fact, differences between HCC patient and healthy individuals are entirely ascribed to extracellular vesicles (EVs), as clearly stated by the title and in the paper itself.
EVs are separated from human sera using exoquick kit. Unfortunately it is well known from the literature that this separation kit suffers from poor selectivity, since many contaminants co-precipitate with EVs (including protein aggregates, lipoproteins, and cell’s organelles), and it has to be coupled to other separation strategies in order to obtain pure EVs preparations:
- http://dx.doi.org/10.3402/jev.v3.24858
- https://doi.org/10.2337/db17-1587
Since no analytical technique has been applied to characterize EV-containing sample, and due to well-known drawbacks of the separation technique chosen for this work, I am not fully convinced that the diagnostic potential of the described technique can be attributed to only EVs and not to other contaminants.
For this reason I recommend to provide a characterization of isolated EVs, as suggested by MISEV guidelines, including nanoparticle tracking analysis, TEM microscopy and Western Blot of EV related proteins and contaminants.
Reviewer 2 Report
In this paper, the authors evaluated the potential of FTIR spectroscopy of circulating EVs for cancer liquid biopsy. This work is novel and appreciate the authors for the good presentation of the article. I would recommend this article for publication after addressing the following queries.
- The paper title includes "liquid biopsy", but there are no details in the introduction. I recommend starting the introduction with a liquid biopsy and then introduce EVs.
- For readers convenience, briefly explain PCA, PCA-LDA and LOOCV.
- The specificity for the classification is low. Have you tried other classification models like PLS-DA?. Why the authors used only PCA-LDA for this study?. Please add a sentence regarding it in the data analysis section.
- Sample size and classification specificity is low. I hope authors will extend this study on larger sample size to validate the obtained results.
Round 2
Reviewer 1 Report
The authors addressed all my comments and suggestions. Now the paper can be published in the present form.